# Effects of Caffeine and Glucose Supplementation at Birth on Piglet Pre-Weaning Growth, Thermoregulation, and Survival

**DOI:** 10.3390/ani13030435

**Published:** 2023-01-27

**Authors:** Lillie Jarratt, Sarah E. James, Roy N. Kirkwood, Tanya L. Nowland

**Affiliations:** 1School of Animal and Veterinary Sciences, The University of Adelaide, Roseworthy, SA 5371, Australia; 2Aquatic and Livestock Sciences, South Australian Research and Development Institute, The University of Adelaide, Roseworthy, SA 5371, Australia

**Keywords:** piglets, neonate, colostrum intake, caffeine, glucose, pre-weaning survival

## Abstract

**Simple Summary:**

Genetic selection for larger litters has been a predominant driver for production efficiency improvements in the pork industry. However, the exacerbation of pre-weaning mortalities occurred simultaneously due to uterine crowding and extended farrowing durations, producing less viable piglets. Supplementing newborn piglets with caffeine has previously been shown to have potential for improving the survivability of lower viable piglets by providing neuroprotection. However, caffeine increases energy utilisation so is potentially counterproductive. To counter a possible energy limitation, our study aimed to investigate whether the addition of glucose to a caffeine supplement would improve piglet vigour and encourage quicker milk acquisition for growth and thermoregulation. We found that caffeine and glucose administered together improved early life growth of low birth weight piglets.

**Abstract:**

Piglet pre-weaning mortality of approximately 15% represents a major economic and welfare concern to the pork industry. Supplementing neonatal piglets with glucose and/or caffeine has the potential to counteract hypoxic stress experienced during parturition and provide an energy substrate, which may improve survival to weaning. This study investigated the effects of caffeine and glucose supplementation at birth, in combination or separately, on piglet growth, thermoregulatory ability, and pre-weaning survival. At birth, 398 piglets were assigned to one of four oral treatments: saline, glucose (300 mg), caffeine (30 mg), or caffeine and glucose combined (30 mg caffeine and 300 mg glucose), dissolved in 6 mL saline. Piglets were tagged at birth, and time taken to reach the udder was recorded. Rectal temperatures were recorded at 4 h and 24 h post-partum, and body weights recorded at birth and 1, 3, and 18 days of age. Colostrum intake was estimated using birth and day 1 weights, and all pre-weaning mortalities were recorded. Treatments did not affect rectal temperature, colostrum intake, or pre-weaning mortality (*p* > 0.05). Low birth weight piglets (<0.9 kg) treated with caffeine and glucose had increased growth between 1 and 3 days of age (*p* < 0.05) compared to low birth weight piglets of other treatment groups. Caffeine supplementation alone reduced overall pre-weaning growth in low birth weight piglets compared to all other treatments (*p* = 0.05). Oral caffeine and glucose had no significant effect on piglet performance except in low birthweight piglets, where it improved growth in the first 3 days of life. Caffeine and glucose supplementation in combination may be beneficial for low birth weight piglets.

## 1. Introduction

Selection for larger litter sizes, targeted to improve farming efficiency, has exacerbated the issue of pre-weaning mortality in the pig industry [1]. Pre-weaning mortality presents both economic and welfare concerns, accounting for 11–20% of piglet deaths [2]. Of these pre-weaning mortalities, 70–90% occur within the first three days postpartum [3]. The main causes are crushing by sows (up to 58%) and low piglet viability (2–30%) [4]. Larger litter sizes lead to an increased in litter weight variation, the proportion of less viable piglets as birth weights decrease, and the number of intrauterine growth restricted piglets. In addition to this, increases in parturition duration result in a higher risk of piglet hypoxia, causing foetal acidaemia and decreased piglet vitality in the first 72 h from birth [5]. Hypoxic pigs often have a lower chance of survival because, in part, they take longer to reach the udder and are less competitive once there. This negatively impacts colostrum intake and reduces immunity due to a lower acquisition of energy and immunoglobulins, decreasing the chance of survival, particularly in low birth weight piglets [6]. The addressing of viability issues of newborn piglets during their most vulnerable time is the appropriate time to target improvements in pre-weaning mortality.

Caffeine, a methylxanthine, has been shown to improve neurological impairment and provide neuroprotection to neonates via its ability to increase pulmonary carbon dioxide sensitivity, increase metabolic rate (increase heart rate), and increase the contractility of the diaphragm, as well as decrease muscle fatigue and diuresis [7]. Caffeine supplemented to rat and mice neonates have been shown to improve myelination defects and reduce brain injuries by increasing oxygen delivery to the brain, which is essential for preventing post-natal parturition-induced hypoxic deaths [8,9]. Studies investigating the efficacy of caffeine supplementation to both low and higher birth weight piglets documented improved metabolic variables of higher birth weight neonates, such as triglyceride, lactate, and blood glucose concentrations, as well as the increased growth of heavier weight piglets, suggesting that caffeine influenced energy utilisation [6,10]. Interestingly, Nowland et al. [10] administered caffeine at birth and 24 h and observed an increase in mortalities of low birth weight piglets, suggesting it to be a result of these piglets having insufficient body energy stores to withstand the increased energy utilisation caused by caffeine supplementation, which, consequently, instead induced a hypoglycaemic state. If this suggestion is accepted, it is plausible that the concurrent addition of glucose to counter the risk of hypoglycaemia may be a potential solution to prevent this issue. Glucose injections have previously been shown to increase blood glucose concentrations in the first 14 h of life and have been associated with an increased piglet body weight during their first 21 d [11]. 

Therefore, the present study aimed to determine whether supplementing caffeine and/or glucose to piglets at birth is beneficial for piglet viability, growth, thermoregulatory ability, and survival. It was hypothesised that the combined supplementation of glucose and caffeine to piglets at birth will decrease piglet pre-weaning mortality and improve their growth and thermoregulatory ability.

## 2. Materials and Methods

### 2.1. Animal Management

This research was conducted at the University of Adelaide’s Roseworthy Piggery, South Australia, in accordance with the Australian Code for the care and use of animals for scientific purposes (8th edition, 2013), and was approved by the University of Adelaide ethics committee (Application ID 35136). The study was conducted in two replicates in August and September 2021. In total, 38 multiparous (parity 1–7, average 3.15 ± 1.63) Large White x Landrace sows and their litters were used (total born; n = 12.92 ± 0.53, born alive; n = 11.92 ± 0.49, stillborn; n = 1.00 ± 0.21). Sows were group housed during gestation and were moved into conventional farrowing crates 4.6 ± 0.05 days prior to their expected due date. Farrowing crates were 1.7 × 2.4 m with slatted flooring within temperature-controlled rooms that were maintained at 22 °C prior to farrowing and throughout lactation. Each farrowing crate had a trough feeder and nipple drinkers for the sow and a heated creep area for piglets on one side. Sows were fed a commercial lactation diet formulated to provide 14 MJ DE/kg, 17.0% crude protein, 0.81% total lysine, and 5% crude fibre, with ad libitum access to water. Sows were fed 2.5 kg/day until farrowing and then gradually increased to reach a maximum of 7–8 kg/d by day 7 of lactation, split between a morning and afternoon feed.

Farrowings were induced on day 113 of gestation with split-dose cloprostenol (Juramate®, Jurox Pty, Ltd., Rutherford, NSW, Australia; 250 µg/mL) vulval injections (0.5 mL at 07:00 and 15:00) to induce farrowing during staffed hours. Sows were monitored throughout farrowing, and manual interventions performed if birth intervals exceeded 2 h between each of the first three piglets born or after one hour from the fourth piglet onwards. No trial sows required extra assistance, such as oxytocin administration, above this protocol. Minimal intervention was provided to piglets post-partum. Cross-fostering was performed, based on litter size and teat capacity (12 ± 2), within 48 h of birth. The average litter size post foster was 11.99 ± 0.05, and piglets were weaned at 21 ± 0.5 days of age.

### 2.2. Experimental Design

At birth, a total of 398 piglets were tagged for individual identification, weighed, their sex recorded, and given one of the following oral treatments:

6 mL saline (SAL; n = 101);

6 mL saline with 30 mg caffeine (CAFF, n = 98);

6 mL saline with 300 mg glucose (GLUC, n = 101);

6 mL saline with 30 mg caffeine and 300 mg glucose (GLUC/CAFF, n = 98).

Treatment allocation was predetermined prenatally for each litter and spread evenly across birth order to achieve an even balance across the four treatments. A minimum of four piglets from each litter were treated to ensure each treatment was represented. Treatments were pre-warmed to 36–40 °C to minimise the impact on piglet body temperature and were orally administered to piglets by syringe. Once treated, all piglets were ear tagged for individual identification and then placed back into the farrowing crate behind the sow to minimise any potential influence on time taken to reach the udder.

### 2.3. Piglet Measurements

Data recorded for each litter were total number of piglets born, born alive, or stillborn and farrowing duration. Piglet rectal temperatures were recorded at 4 and 24 h after birth. All piglets were weighed at birth and again at 1, 3, and 18 days of age. Colostrum intake was estimated from birth and 24 h bodyweights using the equation of Devillers et al. [12]. Video footage was recorded for a subset of 24 litters and later analysed to measure the time taken for piglets to reach the udder following birth, indicated by nose contact with a teat. All piglet fosters and mortalities, including date and reason, were recorded from farrowing to weaning.

### 2.4. Statistical Analysis

Data analysis was performed using the IBM SPSS statistical package, version 25. Data were assessed for normality and outliers, with transformations of the data implemented where necessary (logarithmic transformations of time to udder and colostrum intake data, and square root transformation for 24 h rectal temperature data). Results were considered statistically significant if *p* < 0.05. A linear mixed model was used to assess and compare treatment effects on piglet growth (kg), body weight (kg), temperature (°C), and colostrum intake (g), along with the behavioural measure of time to udder (min). The fixed effects were treatment (SAL, CAFF, GLUC, or CAFF-GLUC), replicate (1 or 2), room (1, 4, and 5), sex, sow parity group (Group 1: parity 1–3 or Group 2: parity ≥ 4), birth order group, and birth weight category (light: <0.9 kg, medium: 1.0–1.4 kg or heavy: >1.5 kg). Birth sow was fitted as a random term. All two-way interactions were assessed and removed from the model if not significant. A non-parametric Kruskal–Wallis test was used to investigate piglet mortality data. Data are presented as estimated marginal mean ± standard error of the mean (SEM).

## 3. Results

There were no significant differences among treatments for the time taken for piglets to reach the udder following birth, rectal temperatures at 4 and 24 h, or colostrum intake (*p* > 0.05; Table 1). Additionally, no significant differences between treatments for weight at birth or 1 and 18 days of age were observed (*p* > 0.05; Table 2). However, at 3 days of age, piglets in the saline treatment group were significantly heavier than all other treatments (*p* < 0.05; Table 2).

When evaluating piglet growth and performance, interactions between treatment and birth weight category were observed. Piglets born light grew significantly less between 1 and 3 d of age when supplemented with caffeine or glucose at birth (*p* = 0.004; Figure 1). However, treatment had no effect on medium weight pigs, while those born heavy grew significantly faster if they were supplemented with glucose or saline (Figure 1). When evaluating the interaction of treatment with birth weight category for growth between 1 and 18 days of age, piglets born light and treated with caffeine at birth grew significantly less than all other treatment groups, while medium and heavy piglets were not significantly affected by any of the birth treatments (Figure 2).

Overall, treatment did not negatively or positively influence piglet pre-weaning mortality (Table 3). The majority of mortalities occurred in the first 3 days (8.1%), with the least number of mortalities occurring between day 3 and 18 (3.2%; Table 3).

## 4. Discussion

This study investigated the effects of providing glucose and/or caffeine to piglets at birth on factors important for piglet survival. At birth, piglets have relatively low body energy reserves for thermoregulation. Thereafter, litter competition at the udder and additional stressors such as increased litter size and lower colostrum intakes limit the available energy supply and so can adversely affect their survival [6]. In addition, Herpin et al. [13] showed that hypoxic piglets, such as those associated with a prolonged farrowing [14], have an impaired glucose uptake and utilisation in peripheral tissues due to the activation of their sympathetic nervous system. The risk of piglet mortality is highest during the first 3 days of life, especially for low birthweight piglets who have relatively low energy reserves [3,15]. Clearly, effective interventions are needed to support an adequate colostrum intake and pre-weaning survival of these neonatal low birthweight piglets. The approach taken in the present study was a bolus of oral glucose and/or caffeine, since we reasoned that low birthweight piglets have a greater surface area to volume ratio and so a greater propensity to lose heat, which would require energy to correct. In the absence of an adequate colostrum intake, this energy must be supplied by other ways. Studies have also demonstrated that compared to other neonates, piglets are at an increased risk of hypothermia as they are born without brown adipose tissue stores [16]. As these energy stores serve as insulators and are readily utilised to produce heat and raise core body temperature, small piglets are at an added disadvantage [17]. Hence, the addition of oral glucose was hoped to aid the piglet’s thermoregulatory ability. However, we detected no treatment effect on piglet rectal temperatures at 4 and 24 h of age, possibly because a single caffeine and/or glucose bolus would have only a relatively short-term effect on blood glucose content and piglet thermoregulatory ability. 

In addition to a potentially impaired energy status, smaller piglets are physically less competitive at the udder. With this in mind, we reasoned that a metabolic stimulus, such as provided by caffeine, would provide a short-term energy boost with consequent improved access to colostrum and ongoing competitiveness. However, a previous study noted that when administered in isolation, caffeine compromised piglet survival [10], presumably via an increased metabolic activity and energy expenditure resulting in hypoglycaemia. To counter this possibility, we included glucose with the caffeine, but this was without effect, again, possibly as a consequence of the short term physiological effect of the glucose. We also noted that the caffeine-treated piglets had impaired weight gains, especially in low birthweight piglets. Presumably, this reflects impaired colostrum and subsequent milk intakes. Our ultimate goal was to increase piglet colostrum consumption and success in accessing ongoing milk intakes. Clearly, this was not evident from our data.

Another possible advantage of caffeine is its influence as an adenosine receptor antagonist, potentially providing extra support to those animals that had suffered hypoxic distress during parturition [8,18]. Studies in humans have demonstrated that caffeine administration to premature neonates improved cardiorespiratory effects and reduced the occurrence of apnea [19]. It is important to note that within these studies the participants were very likely given colostrum or formula during and after caffeine administration, and so negating the negative effects we have observed for weight gain of piglets within the current study and pre-weaning survival in a previous study [10]. 

To our knowledge, this is the first study to investigate the effect of glucose and caffeine in combination, aiming to supply low viable piglets with a readily available energy source while using the action of caffeine to increase their metabolism and energy utilisation. Interestingly, Orozco-Gregorio et al. [6] showed that caffeine has the ability to increase circulating glucose concentration by increasing gluconeogenesis. We hypothesised that the previously noted hypoglycaemic state induced by caffeine would be compensated for by the addition of a glucose supplement when given in combination. In partial support of this, we did note an increased weight gain in low birthweight piglets in response to the combined caffeine/glucose supplement. 

Engelsmann et al. [11] and others [17] noted that the ability to maintain body temperature is a key aspect to piglet survival and that warming of piglets has key benefits to their survival. This suggests that the warming of solutions prior to administration could also have a benefit to piglets at birth and aid in thermoregulation. Indeed, it has been demonstrated that an intraperitoneal injection of warm saline markedly increased colostrum intake and subsequent survival of low birthweight piglets [20]. However, it was not possible in that study to separate the effects of warming per se and an increased colostrum intake on piglet survival. Furthermore, the influence of saline on electrolyte balance cannot be ignored; however, although not relevant to the present study, it reinforces the concept that pre-weaning piglet survival is multifactorial, with utilisation of energy and factors leading to low birthweight and/or viability in the external environment being key factors. Future research could examine combining glucose with colostrum to provide for both relatively immediate and ongoing energy support for caffeine-treated piglets on their growth and survival.

## 5. Conclusions

The improvement of pre-weaning piglet survival will require a focus on the low birth weight/lower viability piglets. Our data suggests that the combination of caffeine and glucose has the potential to improve neonatal performance but that both a rapid and a more prolonged energy support, in addition to improvements in piglet warming, is likely needed for greater effect.

## Figures and Tables

**Figure 1 animals-13-00435-f001:**
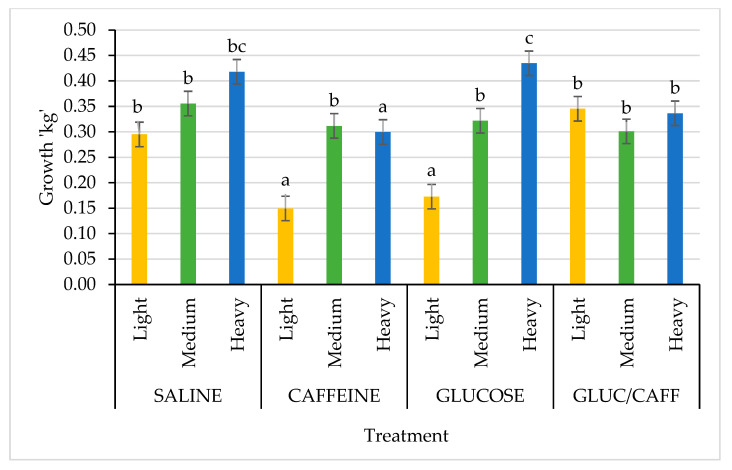
Comparison of growth for birth weight categories (light (yellow): <0.9 kg, medium (green): 1–1.4 kg, heavy (blue): >1.5 kg) from day one to day three of age between the four treatments: saline (SAL), caffeine (CAFF), glucose (GLUC), and caffeine plus glucose (CAFF-GLUC). Means with differing letters (abc) are significantly different (*p* < 0.05). Data shown as mean ± SEM (error bars).

**Figure 2 animals-13-00435-f002:**
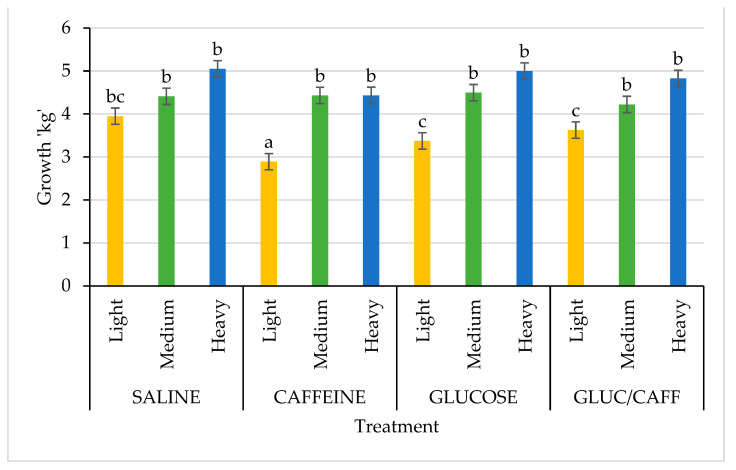
Comparison of growth for birth weight categories (light (yellow): <0.9 kg, medium (green): 1–1.4 kg, heavy (blue): >1.5 kg) from day 1 to day 18 of age between the four treatments: saline (SAL), caffeine (CAFF), glucose (GLUC), and caffeine plus glucose (CAFF-GLUC). Means with differing letters (abc) are significantly different (*p* < 0.05). Data shown as mean ± SEM (error bars).

**Table 1 animals-13-00435-t001:** Rectal temperatures at 4 and 24 h (°C), time to udder (min), and colostrum intakes (g) for piglets allocated to oral treatments at birth groups; saline (SAL), caffeine (CAFF), glucose (GLUC) and caffeine plus glucose (CAFF-GLUC).

Variables	n	SAL	CAFF	GLUC	CAFF-GLUC	*p*-Value
4 h temperature (°C)	218	36.33 ± 0.25	36.61 ± 0.26	36.49 ± 0.27	36.61 ± 0.25	0.705
24 h temperature (°C) #	355	6.16 ± 0.10	6.15 ± 0.01	6.15 ± 0.01	6.14 ± 0.01	0.566
Time to udder (min) *	263	1.24 ± 0.07	1.16 ± 0.80	1.21 ± 0.08	1.21 ± 0.08	0.560
Colostrum intake (g) *	340	2.44 ± 0.30	2.48 ± 0.04	2.47 ± 0.03	2.48 ± 0.30	0.677

* Log transformed data; # sqrt transformed data. n represents the number of animals sampled for each measurement. Data shown as estimated marginal mean ± SEM.

**Table 2 animals-13-00435-t002:** Body weights (kg) at birth, 1, 3 and 18 days of age, for piglets allocated to oral treatments at birth groups; saline (SAL), caffeine (CAFF), glucose (GLUC) and caffeine plus glucose (CAFF-GLUC).

Variables	n	SAL	CAFF	GLUC	CAFF-GLUC	*p*-Value
Day 0 weight (kg)	392	1.42 ± 0.02	1.40 ± 0.02	1.38 ± 0.02	1.42 ± 0.02	0.221
Day 1 weight (kg)	352	1.56 ± 0.02	1.51 ± 0.03	1.50 ± 0.02	1.53 ± 0.02	0.195
Day 3 weight (kg)	334	1.93 ± 0.03 ^a^	1.81 ± 0.03 ^b^	1.81 ± 0.03 ^b^	1.86 ± 0.03 ^b^	0.003
Day 18 weight (kg)	324	5.88 ± 0.15	5.60 ± 0.15	5.83 ± 0.15	5.67 ± 0.15	0.131

Values with different superscripts differ significantly (*p* < 0.05). n represents the number of animals sampled at each weighing time point. Data shown as estimated marginal mean ± SEM.

**Table 3 animals-13-00435-t003:** Mortality at day 1, 3, and 18 of piglets allocated to oral treatments at birth; saline (SAL), caffeine (CAFF), glucose (GLUC) and caffeine plus glucose (CAFF-GLUC).

Mortality	n	SAL	CAFF	GLUC	CAFF-GLUC	Total	*p*-Value
Day 0 to 1	28	8/101 (7.9%)	6/98 (6.1%)	5/100 (5%)	9/98 (9.2%)	28/397 (7.1%)	0.667
Day 1 to 3	30	7/93 (7.5%)	6/92 (6.5%)	11/95 (11.6%)	6/89 (6.7%)	30/369 (8.1%)	0.551
Day 3 to 18	11	3/86 (3.5%)	6/86 (7%)	1/84 (1.2%)	1/83 (1.2%)	11/339 (3.2%)	0.109
Total	69	18/280 (6.4%)	18/276 (6.5%)	17/279 (6.1%)	16/270 (5.9%)	18.4%	0.983

n represents the number of animals at each time point.

## Data Availability

The data supporting this trial will be shared on reasonable request to the corresponding author.

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
