# Peer review of "Effects of Caffeine and Glucose Supplementation at Birth on Piglet Pre-Weaning Growth, Thermoregulation, and Survival"

_animals, 2023, doi:10.3390/ani13030435_

Round 1
Reviewer 1 Report
The main research objective of the manuscript was to explore the effects of oral caffeine or glucose or caffeine and glucose on pre-weaning growth, thermoregulation, and survival in low vigor piglets. Studies have shown that the combination of caffeine and glucose has some potential for the growth and survival rate of piglets before weaning. The results are of guiding significance for the management of piglets before weaning. Overall, this article is well-structured and logically clear. This manuscript is intended for publication in the Animals, with minor revisions. Specific comments/suggestions are as follows.
1. Line 30:Please change "Treatment "to "Treatments".
2. Line 31:Please italicize the P value and modify it throughout the text.
3. Line 32:Please represent significant P values as P < 0.05.
4. Line 38:Change "Pig" to "piglets" and do not need to capitalize the first letter.
5. Line 38: Modify "preweaning survival" to "pre-weaning survival".
6. Line 129:Please change "treatment "to "treatments".
7. Line 138:Use italics for significant P values.
8. Line 141:For significant values, use p < 0.05.
9. Line 145-150: Please explain the n values in Table 1 and Table 2.
10. Line 150: Use italics for significant P values.
11. Line 154:Use italics for significant P values.
12. Line 162-164,166-169:Please note the space between the symbol and the number.
13. Line 160-161,164-165: Please indicate when the piglets are weighed in Figure 1 and Figure 2 respectively. In addition, what do 1,2 and 3 represent in the figure respectively?
14. Please explain the n values in Table 3.
Reviewer 2 Report
line 3: I do suggest to remove the "low viable" term from the title, as the research not focused only on low viable piglets, actually we do not know how many of them were.
line 47: Authors needs to discuss low birth weight and IUGR piglets as well in relation to litter size.
line 84: This number of piglets are not high. Modern genotypes can have 18-19 born alive on average. This needs to be discussed. What was the average functional teat numbers of the sows?
line 90: would be more pericise for comparison to provide SID Lys content, and ratios of other amino acids, and level of macro minerals as well.
line 91 :Were the sows fed by hand? What about feed residues? What about actual feed intake?
line 99: cross fostering to what litter size? Maybe the effect of fostering should have been tested statistically. Which piglets were removed (the biggest ones)?
Table 1: In MS Word insert-symbols there is a small circle in upper position, which can be used more elegantly to mark celsius.
Table 2: what data provided after plus-minus sign: standard error or standard deviation? It needs to be given.
Below the table: superscripts maybe better term instead of subscripts.
line 153: the number of piglets in different weight categories over treatments needs to be presented.
Figure 1 and 2: letters over bars needs to be explained. Title inappropriate: birth weight categories are compared, not growth weight categories.
Table 3: n 29 and 12 not matching with the number in the Total column. In the total row summing up the piglets is incorrect, because these are the same piglets. For instance in the first column, the correct calculation is 18/101(17.8%) and so on. The average loss is not correct: 69 / 397 gives 17.4%.
line 214: Saline solution is not without active component: sodium and chloride. Some research demonstrated that piglets are having higher requirement for chloride. Electrolite balance is important. This aspect needs to be discussed. The discussion needs to be expanded in general, the manuscript not well balanced at the moment.
line 237: to the? not "by the"?
